# PD-1 Inhibitors-Related Neurological Toxicities in Patients with Non-Small-Cell Lung Cancer: A Literature Review

**DOI:** 10.3390/cancers11030296

**Published:** 2019-03-01

**Authors:** Aurora Mirabile, Elena Brioschi, Monika Ducceschi, Sheila Piva, Chiara Lazzari, Alessandra Bulotta, Maria Grazia Viganò, Giovanna Petrella, Luca Gianni, Vanesa Gregorc

**Affiliations:** 1Department of Oncology, Division of Experimental Medicine, IRCCS San Raffaele, Via Olgettina 60, 20132 Milan, Italy; brioschi.elena@hsr.it (E.B.); Ducceschi.monika@hsr.it (M.D.); chiara.lazzari@hsr.it (C.L.); bulotta.alessandra@hsr.it (A.B.); vigano.mariagrazia@hsr.it (M.G.V.); petrella.giovanna@hsr.it (G.P.); gianni.luca@hsr.it (L.G.); gregorc.vanesa@hsr.it (V.G.); 2Department of Oncology, ASST Fatebenefratelli Sacco, 20121 Milan, Italy; sheila.piva@asst-fbf-sacco.it

**Keywords:** immunotherapy, neurotoxicity, polyneuropathy, myasthenia gravis, Bell’s palsy, encephalopathy, nivolumab, pembrolizumab

## Abstract

The advent of immune checkpoint inhibitors gave rise to a new era in oncology and general medicine. The increasing use of programmed death-1 (PD-1) inhibitors in non-small cell lung cancer and in other malignancies means clinicians have to face up to new challenges in managing immune-related adverse events (irAEs), which often resemble autoimmune diseases. Neurological irAEs represent an emerging toxicity related to immunotherapy, and it is mandatory to know how to monitor, recognize, and manage them, since they can rapidly lead to patient death if untreated. Guidelines for the diagnosis and treatment of these irAEs have been recently published but sharing some of the most unusual clinical cases is crucial, in our opinion, to improve awareness and to optimize the approach for these patients. A literature review on the diagnosis and treatment of immune-related neurotoxicity’s has been conducted starting from the report of four cases of neurological irAEs regarding cases of polyneuropathy, myasthenia gravis, Bell’s palsy, and encephalopathy, all of which occurred in oncological patients receiving PD-1 inhibitors (pembrolizumab and nivolumab) for the treatment of non-oncogene addicted advanced non-small cell lung cancer. The exclusion of other differential diagnoses and the correlation between the suspension of immunotherapy and improvement of symptoms suggest that immunotherapy could be the cause of the neurological disorders reported.

## 1. Introduction

Worldwide, lung cancer is the most common malignancy and has one of the highest mortality rates [1].

In 2014, the approval by the Food and Drug Administration (FDA) of programmed death-1 (PD-1) inhibitors, pembrolizumab and nivolumab, revolutionized the landscape of non-oncogene addicted stage IV non-small cell lung cancer (NSCLC) treatment. 

Pembrolizumab is a humanized monoclonal antibody directed against the negative immunoregulatory human cell surface receptor programmed death-1 (PD-1) which is effective as an immune checkpoint inhibitor and has antineoplastic activity. 

Nivolumab is a fully human immunoglobulin G4 monoclonal antibody, also directed against PD-1. The activation of T-cells and cell-mediated immune responses against the tumor are enhanced by blocking the activation of PD-1 by its ligands programmed cell death ligand 1 (PD-L1)—overexpressed on certain cancer cells—and programmed cell death ligand 2 (PD-L2), which is primarily expressed on antigen-presenting cells. In fact, activated PD-1 negatively regulates T-cell activation, playing a fundamental role in tumor escape from host immunity. 

The increasing use of these treatments brings new challenges, as clinicians must manage immune-related adverse events, which have never been observed with conventional chemotherapies, and which often resemble autoimmune diseases. 

The most common immune-related adverse events (irAEs) reported in clinical trials among NSCLC patients receiving PD-1 inhibitors include: Autoimmune hypophysitis, thyroiditis, colitis, hepatitis, pneumonitis, and a rash, sometimes appearing as systemic diseases [2]. 

The exact pathophysiology leading to irAEs remains unclear. Several different mechanisms seem to be involved in the development of irAEs rather than a single process. Many irAEs are similar to symptoms we can observe in autoimmune diseases, suggesting that they share mechanisms that lead to failure in self-tolerance [3]. 

The early recognition and treatment of irAEs, even in their subclinical stage, is crucial both for the resolution of symptoms and treatment management. Nevertheless, PD-1 inhibitors-associated irAEs that affect the nervous system are rarely reported and the pathogenesis of neurological irAEs is still unclear. Checkpoint inhibition can precipitate underlying autoimmune disorders, but the data available in the literature are mainly about the neurological side effects of ipilimumab (e.g., ipilimumab can induce and exacerbate myasthenia gravis, a disease caused by T-cell-mediated production of acetylcholine receptor antibodies) and or in patients affected by advanced melanoma. Moreover, paraneoplastic syndromes could provide important clues about which shared neuron-specific antigens could precipitate autoimmunity and induce irAEs [4].

The aim of our manuscript is to review the literature of these uncommon side effects starting from the example of four different cases of PD-1 inhibitors-associated neuro-toxicities (polyneuropathy, myasthenia gravis, Bell’s palsy and encephalopathy) in non-oncogene addicted stage IV NSCLC patients, to better describe the difficulties physicians must deal with.

As the use of these agents increases in other tumor types, it is important for clinicians to be aware of the serious potential side effects, such as immune-related neurological toxicities, which may have lasting consequences. Even if they are rare and often respond well to steroid treatment, they can present in different patterns, and do not always have a favorable outcome. Different specialists’ consultations are necessary in order to classify and successfully treat these conditions, as many patients have a reasonable chance of long-term disease control. 

## 2. Materials and Methods

### 2.1. Case Reports

We retrospectively selected four cases of patients with a known diagnosis of advanced NSCLC treated with immunotherapy from January 2017 to December 2017 with the following inclusion criteria: Histologically diagnosed NSCLC, immunotherapy-related neurotoxicity, and treatment with anti-PDL-1. Immune-related neurotoxicity was defined as a diagnosis of exclusion.

### 2.2. Literature Review

A search of MEDLINE, EMBASE, and CINAHL databases, Cochrane Central Register of Controlled Trial, and the Cochrane Database of Systemic Reviews was done for articles published in English between January 1996 and February 2018. The search terms included immunotherapy toxicity or adverse events, neurotoxicity and cancer treatment, nivolumab or pembrolizumab and neurotoxicity. References cited in the articles obtained from the above search and related articles in MEDLINE were included.

## 3. Case Reports

### 3.1. Case Report 1

The first case is about a 74-year-old man, ECOG PS (Eastern Oncology Cooperative Group Performance Status) 1, diagnosed with non-oncogene addicted lung adenocarcinoma with lung, pleural, bone, and adrenal lesions, with PD-L1 expression in 30% of tumor cells. He progressed to the first line chemotherapy with cisplatin and pemetrexed and underwent second line treatment with nivolumab (3 mg/kg every 2 weeks) in June 2017. Immunotherapy was interrupted in August 2017 after 8 cycles due to disease progression with evidence of spinal infiltration in D3–D6, treated with focused radiotherapy until September 2017.

Two weeks later, he presented with diffuse tremors, difficulty in walking, and head bending. Blood tests excluded other causes such as diabetes, B12 or folate deficiency, thyroid-stimulating hormone (TSH) impairment, and HIV infection. Onconeural antibodies were negative. After a clinical neurological evaluation, an electromyography (EMG) documented a serious axonal motor-sensor polyneuropathy, particularly involving the lower limbs. A spinal computed tomography (CT) and brain Magnetic Resonance Imaging (MRI) did not show signs of myelopathy or metastasis; furthermore, the lung cancer lesions appeared to be stable.

The patient’s syndrome was managed with dexamethasone 16 mg daily with an improvement in neurological symptoms in 4 days and a complete remission in 14 days.

A third line chemotherapy with taxanes was administered at the complete recovery of good clinical conditions.

### 3.2. Case Report 2

In September 2016 a 64-year-old man, ECOG PS 0, was diagnosed with lung, pleural, bone, and brain relapse of a surgically treated non-oncogene addicted lung adenocarcinoma with PD-L1 expression in 10% of tumor cells. He received a gamma-knife treatment on the right frontal and ipsilateral temporal brain lesions and subsequently started first-line chemotherapy with cisplatin plus pemetrexed (4 cycles), followed by maintenance with pemetrexed. 

At disease progression on May 25th 2017, the patient began second line treatment with nivolumab 3 mg/kg every 2 weeks. 

Nine days after the second dose of immunotherapy, the patient developed transaminase elevation and a bilateral medial diplopia. After an evaluation by a neurologist and optician ruled out ocular disorders, migraine, and other cranial nerve disorders, an isolated bilateral sixth cranial nerve deficiency was suspected. No signs of trauma or inflammation were visible. In the suspect of irAES we discontinued immunotherapy.

A brain MRI showed a reduction in the size of the right frontal lesion and a significant reduction of the associated edema, without the appearance of new metastasis or any alterations that might explain the patient’s symptoms and signs. The absence of pain enabled us to rule out orbital myositis and ophthalmoplegic migraine.

Following the immunologist and neurologist’s suggestions, we checked the thyroid function and related autoantibodies and the acetylcholine receptor antibodies. The first were normal. The positivity of the acetylcholine receptor antibodies (AChR) test (1.4 nmol/L, with upper limit of 0.5 nmol/L) as well as the neurologist’s opinion supported the hypothesis of nivolumab-related myasthenia gravis (MG), even though 41% of these cases have negative MG autoantibodies [5]. 

We hospitalized our patient and started methylprednisolone 1 mg/kg, with a quick improvement in neurological symptoms and a progressive reduction of transaminase and AChR levels until complete normalization.

A month after the patient recovery, we resumed immunotherapy that is still ongoing with oncological partial response.

### 3.3. Case Report 3

In March 2016 a 74-year-old man, ECOG PS 1, known for arterial hypertension and carotid stenosis, was diagnosed with non-oncogene addicted stage IV lung adenocarcinoma with PD-L1 expression in 2% of tumor cells. The patient received 4 cycles of first-line chemotherapy with carboplatin plus pemetrexed, followed by 3 cycles of maintenance treatment with pemetrexed. In April 2017, because of disease progression, the patient started second-line treatment with nivolumab 3 mg/kg every 2 weeks. Exactly thirteen days after the first infusion, he developed grade 3 diarrhea without fever or emesis. Suspecting a nivolumab-related colitis, oral methylprednisolone 1 mg/kg was promptly started obtaining a rapid improvement in symptoms and the second dose was delayed. After a week, he started reducing the dose of steroids and on May 15th he resumed nivolumab. 

A few days after the fifth infusion of immunotherapy, while the patient was still tapering the steroid, he developed grade 2 diarrhea and grade 3 asthenia, rapidly followed by mental confusion and dysarthria with evidence of acute isolated left peripheral VII cranial nerve palsy. No electrolyte imbalance, renal function impairment, or signs of dehydration could be detected. 

A brain and facial MRI with gadolinium excluded the presence of brain metastasis, ischemic or hemorrhagic lesions and showed no alterations along the VII cranial nerve. The neurologist diagnosed Bell’s palsy. 

Considering the recurrence of diarrhea at steroid tapering and its association with Bell’s palsy, we suspected that the cause of both symptoms could be immune-related. The diarrhea disappeared after we interrupted the immunotherapy and increased the dose of oral methylprednisolone, but Bell’s palsy remained unchanged. 

In consideration of the patient’s desire to continue treatment, and the good ECOG PS maintained despite neurological toxicity, vinorelbine chemotherapy was started. 

He died in summer 2018 due to disease progression.

### 3.4. Case Report 4

In February 2017 a 55-year-old woman, ECOG PS 1, with a history of arterial hypertension and lung emphysema, received the diagnosis of a non-oncogene addicted stage IV squamous NSCLC. 

In August 2017, when the disease progressed after first line chemotherapy with cisplatin and gemcitabine, she began immunotherapy with pembrolizumab with the schedule of 200 mg every 3 weeks, since PDL-1 was expressed in more than 50% of tumor cells. A few hours after the first dose of immunotherapy, the patient acutely developed mental confusion, drowsiness, and a left brachio-crural motor sparing syndrome. A brain CT scan without contrast excluded acute events and after approximately two hours, neurological symptoms and signs disappeared spontaneously.

In the following days, the same neurological symptoms and signs reappeared intermittently. As suggested by the infectivologist and neurologist, we started empirical treatment with intravenous acyclovir 10 mg/kg every 8 h as in herpetic encephalitis, and an antibiotic treatment with intravenous meropenem 2 gr every 8 h. Further laboratory tests revealed negative IgM and IgG for Herpes Simplex Viruses 1 and 2. 

Assuming that the neurological syndrome could be caused by immunotherapy, we started methylprednisolone at the dose of 2 mg/kg.

Three weeks after the first dose of pembrolizumab the patient showed signs of clinical and neurological improvement with normalization of blood calcium levels, a resolution of leukocytosis, a reduction of C-reactive Protein (CRP), and stable apyrexia. 

We decided to re-challenge pembrolizumab achieving clinical benefit and a very good partial response in the lung and lymph nodes.

## 4. Literature Review and Discussion

Cancer patients treated with immunotherapy, in spite of the promising results, could possibly develop irAEs during treatment with immune checkpoint inhibitors. 

Neurological syndromes occur in 4.2% of cancer patients treated with immunotherapy [6], with an extremely broad spectrum of possible presentations, potentially involving all areas of the central and peripheral nervous system. 

Most of the irAEs described in the literature are related to anti-CTLA4+/− anti PD-1 in melanoma patients [7,8,9]. We found only seven reported cases of anti PD-1 related neurological irAE in NSCLC (Table 1).

At the time of our cases, shared guidelines for the management of irAEs had not yet been published, but we based our therapeutic approach on previous publications and on neurologist’s suggestions.

Specific guidelines for the diagnosis and treatment of neurological irAEs suggested some algorithms [14,15].

Current medical literature showed clinical improvement only after immunotherapy was interrupted and adequate steroid treatment administered as summarized in Figure 1 [16,17].

The following literature review has been made starting from the examples of four cases of neurological adverse events occurring in patients treated with Pembrolizumab or Nivolumab, confirming the broad spectrum of possible clinical presentations. 

Cuzzubbo et al. [18] performed a systematic literature review (up to February 2016) on irEAs in different types of neoplasia, but all related to anti-CTLA4+/− anti PD-1. They found an extremely broad spectrum of possible syndromes, potentially involving all areas of the central and peripheral nervous system. The median time of onset of these events was six weeks. In a few cases, patients were subjected to targeted biopsies and the histological analysis demonstrated a lymphocytic infiltrate, thus supporting an autoimmune involvement. Headache is the most common neurological irAE, both in low and high Common Terminology Criteria for Adverse Events (CTCAE) grades of presentation, followed, in those cases of grade 3 to 4 toxicities, by encephalopathies and meningitis. 

In our cases, the timing of the onset of neurologic symptoms after administration of nivolumab and pembrolizumab suggested immune-related adverse events rather than classic paraneoplastic neurologic disorders. Most tumor-induced paraneoplastic neurologic disorders are subacute and progressive, commonly preceding the detection of the tumor by months or years [4,5,6,7,8,9,14,15,16,17,18,19]. All patients were treated for metastatic cancer for extended periods without evidence of paraneoplastic neurologic disorders.

Finally, all patients had marked clinical improvement after immunosuppressive treatment. 

These features suggest that immune checkpoint inhibition favored the development of immune responses against neuronal antigens.

Peripheral neuropathy is defined as an asymmetric or symmetric sensory, motor, or sensory motor deficit sometimes associated with focal mononeuropathies (Table 2), including cranial neuropathies (e.g., facial neuropathies/Bell’s palsy), numbness and painful or painless paresthesia, hypo/areflexia or sensory ataxia [15]. Immunotherapy-related polyneuropathies are difficult to differentiate, e.g., toxicity due to chemotherapy or to paraneoplastic syndromes. However, among neurological paraneoplastic syndromes, polyneuropathies are classically sensory, asymmetrical or multifocal, frequently involving the upper limbs and characterized by a subacute onset and a rapidly progressive disease course with paresthesia and early pain [19]. 

Peripheral polyneuropathy is another relatively recurrent neurological irAE, with a frequency of 5% of grade 1 to 2 and 6% of grade 3 to 4 neurological adverse events related to immune checkpoints inhibitors. 

As suggested in recently published guidelines on the diagnosis and management of irAEs, we have to consider evaluating patients for causes of reversible neuropathy such as diabetes, B12 or folate deficiency, TSH impairment, and HIV during the consultation with the neurologist, as well as a MRI of spine or of the brain if cranial nerves are involved, and an electromyography. While the management includes prednisone 0.5 to 1 mg/kg (if grade 2) or admission to the ward and therapy with IV methylprednisolone 2 to 4 mg/kg (if grade 3) in addition to neurontin, pregabalin, or duloxetine for the pain, and immunotherapy discontinuation until recovery at least to grade 1 [15].

Moreover, onconeural antibodies (e.g., anti-Hu and anti-CV2/CRMP5) are absent in neuropathy related to immune checkpoint inhibitors and present in 80% of paraneoplastic sensory neuropathies, suggesting a way of developing a differential diagnosis together with electromyography [20].

In our case, immunotherapy was started 9 weeks before and interrupted one month before the radiotherapy was given to the spinal column and this could have favored a previous nivolumab-triggered attack of the immune system against the spinal cord. In fact, according to the theory, irradiated tumor cell death can enhance antitumor immunity by inducing antigen expression on tumor cells and activating lymphocytes [21]. In our case, we probably had a Grade 2 irAE, since blood tests excluded other causes such as diabetes, B12 or folate deficiency, TSH impairment, and HIV infection and negative onconeural antibodies, as well as brain/spinal cord magnetic resonance (MRI) and chest/abdomen CT scan helping us to confirm the suspicion of irAE. Current guidelines suggest to hold immunotherapy and resume once there is a return to G1, initial observation, or initiate prednisone 0.5–1 mg/kg associated to neurontin, pregabalin, or duloxetine for pain [15].

Myastenia gravis (MG) is another emerging toxicity of immune checkpoint inhibitors, with 30.4% of related mortality [16]. MG is defined as fatigability or fluctuating muscle weakness, generally more proximal than distal, frequently associated with ocular and/or bulbar involvement (ptosis, extraocular movement abnormalities resulting in double vision, dysphagia, dysarthria, facial muscle weakness) and sometimes also with neck and/or respiratory muscle weakness (Table 1). It may have overlapping symptoms with the Miller Fisher variant of Guillain-Barre’syndrome (ophthalmoparesis) and oculobulbar myositis (ptosis, ophthalmoparesis, dysphagia, neck and respiratory weakness) [15].

The clinical features of immune-related MG seem to be different from classical MG, particularly in the risk of developing myasthenic crisis, which seem to be more frequent and appear earlier [21], occurring generally within 1 to 4 cycles of immunotherapy.

Literature data report cases where ipilimumab was the main cause or was administered as part of the oncological treatment in association with nivolumab. 

In a case of nivolumab-induced MG reported by Hasegawa et al. [10], the patient presented also with creatine phosphokinase (CPK) and transaminase elevation. In our case, a central nervous system MRI showed a dimensional reduction of the brain metastasis and of the associated edema, without any alterations that might explain the patient’s symptoms and signs and acetylcholine receptor antibodies were normal. This confirmed the hypothesis of irAEs even if, according to the recent guidelines, acetylcholine receptor antibodies were negative, testing for muscle-specific kinase and the four lipoprotein-related antibodies should be considered [15].

In our case, monotherapy with nivolumab was sufficient to trigger a Grade 2 MG and, as described by Hasegawa et al. [10], neurological symptoms were associated with transaminase elevation. The positivity of acetylcholine receptor antibodies and the neurological evaluation were essential to confirm the immune-related hypothesis [4]. 

Current guidelines suggest to hold immunotherapy and resuming in G2 patients only if symptoms resolve, consulting neurology, starting pyridostigmine at 30 mg orally three times a day and gradually increasing to a maximum of 120 mg orally four times a day as tolerated and based on symptoms, and administer corticosteroids (prednisone, 1–1.5 mg/kg orally daily) if symptoms G2 [15].

We started with methylprednisolone 1 mg/kg, obtaining a quick improvement in neurological symptoms and a progressive reduction of transaminase and AChR levels until complete normalization, thus that we did not need to add other treatments.

The mechanism of MG induction is unclear, but it is suspected to be due to cytotoxic T-lymphocyte activation.

In 2017, Suzuki et al., worked on a 2-year safety database, based on post-marketing surveys in Japan, in order to report the clinical features of MG induced by treatment with immune checkpoint inhibitors.

He compared 10,277 patients treated with either nivolumab or ipilimumab to 105 patients with idiopathic MG.

Six of 12 patients with nivolumab-related myasthenia gravis were affected by NSCLC.

Bulbar symptoms and myasthenic crisis were observed more frequently than in idiopathic MG in nivolumab-related MG. Ten patients were positive for acetylcholine receptor antibodies. Immunosuppressive therapy was effective in patients with nivolumab-related MG. Mild symptoms responded to oral prednisolone plus pyridostigmine, and patients recovered within several weeks. Intravenous methylprednisolone, immunoglobulins, and plasma exchange therapy were combined to reduce the rapid progression of MG in patients with severe involvement. A gradual improvement in muscle strength showed over 4–8 weeks in patients with severe involvement [22].

The prompt and correct recognition of MG following treatment with immune checkpoint inhibitors in patients with cancer is important to reduce admission time and to obtain rapid improvement of symptoms.

The third case presented an extremely rare neurological irAE: Bell’s palsy. 

It is defined as a temporary weakness or lack of movement affecting one side of the face and it is the most common peripheral paralysis of the seventh cranial nerve with an onset that is rapid and unilateral (Table 2). 

The diagnosis is of exclusion and it is most often made based on a physical examination [23].

There are only six cases reported in the literature, which describe the development of unilateral or bilateral facial palsy in melanoma patients treated with immune checkpoint inhibitors (one case with ipilimumab, four cases with the combination of ipilimumab and nivolumab, and one with the combination of ipilimumab and pembrolizumab), which responded well to steroid administration [11,24,25].

The exact cause of Bell’s palsy remains unclear, but various etiologies have been suggested, such as viral, inflammatory, autoimmune, and vascular. The most likely cause is suspected to be the reactivation of herpes simplex virus or herpes zoster virus from the geniculate ganglion [26,27]. 

It is clear that inflammation and edema of the facial nerve were responsible for the symptoms.

No guidelines are currently available for the management and treatment of immune-related Bell’s palsy. 

Corticosteroids are used for their anti-inflammatory effect and are most effective when started within 72 h from the onset of symptoms. In adults, daily doses of 50–60 mg prednisolone continued for 10 days have been commonly used [27,28]. A Cochrane review in 2015 found that the combination of steroids plus antivirals reduced long-term sequelae such as excessive tear production and synkinesis and that the outcome for patients who received corticosteroids alone was significantly better compared to those who received antivirals alone [16]. 

In our patient, the only possible underlying mechanism is autoimmune as in the other irAEs, suggesting the use of steroids alone. 

As we can see from the cases we described, the diagnosis of neurological irAEs is difficult and is often a diagnosis of exclusion, which requires the cooperation of different specialists. Moreover, treatment needs to be started quickly because symptoms can get worse in a very short time compared to similar not immune-related disorders and can easily lead to patient death. 

Finally, our last case was about one of the most common irAEs: Encephalitis.

As summarized in Table 2 it is defined by the following symptoms: Confusion, altered behavior, headaches, seizures, short-term memory loss, depressed levels of consciousness, focal weakness, speech abnormalities, which are not caused by infections, especially viral infections (e.g., herpes simplex virus) [15].

A headache is the most common neurological irAE, both in low and high CTCAE grades of presentation, followed in those cases of grade 3 to 4 toxicities by encephalopathies and meningitis [14,18].

With most symptoms being nonspecific, a notable risk factor for suspecting an immune checkpoint inhibitor-induced encephalopathy is the short interval between symptom onset and the administration of immunotherapy. In many cases a brain MRI will result negative, while almost constantly an elevated protein level associated with lymphocytic pleocytosis are found in cerebrospinal fluid (CSF) examination [14]. 

In our patient, we did not find an elevated protein level nor lymphocytic pleocytosis in CSF examination. In addition, all the microbiological assays and virus serology were negative in CSF itself and her condition improved with steroids, thus the most probable hypothesis in our case was a Grade 2 irAE. 

In patients treated with immune checkpoint inhibitors, we should always keep in mind the possibility of a correlated encephalopathy [14]. 

More than 10 cases of immune-related encephalitis have been published in the literature. The most frequent presenting complaints have been cognitive impairment, memory loss, and gait disturbance [3].

Data in the literature describe how immune checkpoint inhibitor-induced encephalopathies occur in 19% of treated patients and encephalitis in 1–3% of treated cases, presenting with nonspecific signs of confusion, autonomic instability, waxing and waning mental status, and a negative workup [17,18,29].

In a case presented by Salam et al. a patient developed subacute memory loss without other neurological deficits after 12 months of pembrolizumab. A brain MRI revealed T2 signal changes within the hippocampus and anterior temporal lobe and insula, and analysis of CSF showed proteinemia and lymphocytosis. The results of an electroencephalogram and a paraneoplastic antibody panel were normal. Despite treatment with methylprednisolone administered intravenously, no neurological improvement was seen [7].

In many cases the brain MRI results are negative, while there are almost constantly elevated protein levels and lymphocytic pleocytosis in CSF examination. 

Nevertheless, irAEs typically take place in those organs or tissues most involved in neoplastic disease or in paraneoplastic syndromes where there is greater inflammation. Thus, in patients treated with immune checkpoint inhibitors, the possibility of a correlated encephalopathy should always be kept in mind [18].

In 2016, Williams et al. published two cases of immune-related encephalitis in a NSCLC patient and in a melanoma patient treated with anti PD-1 and anti PDL-1 therapy, respectively. Symptoms disappeared in the first case after empirical high-dose intravenous methylprednisolone sodium succinate equivalent to 1000 mg/day of methylprednisolone for five days followed by 0.4 mg/kg/day of intravenous immunoglobulin for five days, and two doses of Rituximab. In the second case symptoms disappeared after the patient received oral prednisone at a dose of 60 mg/die and the interruption of immunotherapy [4].

A brain MRI with or without contrast could help differentiate autoimmune encephalopathies from limbic encephalitis. A lumbar puncture could help distinguish between a viral infection, autoimmune encephalopathy and paraneoplastic syndromes. An electroencephalography (EEG) could be useful to evaluate subclinical seizures. Moreover, it is necessary to exclude aseptic meningitis, autoimmune syndromes, anemia, and thrombotic thrombocytopenic purpura (TTP) as causes of encephalopathy. The first treatment indicated in current guidelines is methylprednisolone 1 to 2 mg/kg, and, if severe or progressing symptoms or oligoclonal bands present, pulse corticosteroids methylprednisolone 1 g IV daily for 3–5 days plus IV immunoglobulins 2 g/kg over five days. If positive for autoimmune encephalopathy antibodies, and limited or no improvement, consider rituximab or plasmapheresis in consultation with neurology [15].

In 2017, Feng et al. reported a rare case of encephalopathy which developed as a consequence of immunotherapy with pembrolizumab and was treated with high-dose methylprednisolone administered intravenously at a dose of 1 g daily for three days with a near-complete resolution of his expressive dysphasia and improvement in gait. The patient was discharged on a regimen of 60 mg of prednisolone daily, which was progressively tapered by 10 mg each week [3].

On the other hand, there is evidence that suggests that the development of neurological irAEs could be positive predictors of disease response to immunotherapy.

Spain et al. reported a tumor objective response rate of 70% in patients who experienced neurological irAEs (in contrast to a 20–30% overall response rate in phase III trials), with a median overall survival of 45.7 months compared with 11.2 months in those without neurological irAEs [30]. Other literature data revealed that 40 to 50% of patients who developed neurological irAEs achieved a partial or complete response [9,11,29]. Possibly, these are anecdotal and uncontrolled data, but they suggest that the development of neurological irAEs may be associated with an increase in overall response rate in spite of the use of the high-dose corticosteroids required to treat neurological irAEs. This leaves space for the hypothesis that in these cases, it does not reduce the efficacy of immunotherapy. 

Moreover, despite the required discontinuation of therapy after only a few doses, our patients had a sustained response on imaging and a slight improvement in oncological symptoms.

## 5. Conclusions

This review focused on a growing spectrum of syndromes that we need to recognize, diagnose, and treat properly, especially in this new era where immune checkpoint inhibitors are becoming an essential step in therapeutic strategies in a growing number of tumors. Quiz Ref IDEarly recognition and management of these neurologic irAEs is essential in improving clinical recovery and reducing the effect of drug-related toxicities. It is also necessary to educate health professionals, patients, and their caregivers, thus that they can promptly recognize symptoms and suspicious signs. In particular, in our experience, the cooperation between different specialists was essential to achieve symptom recovery and to allow the prosecution of immunotherapy. Further research is required to study the patterns of neurotoxicity, identify the underlying pathogenesis, and to optimize the treatment paradigm. In addition, an international database, especially for the rarest irAEs, could be helpful for early identification and to improve the management of these complex toxicities.

## Figures and Tables

**Figure 1 cancers-11-00296-f001:**
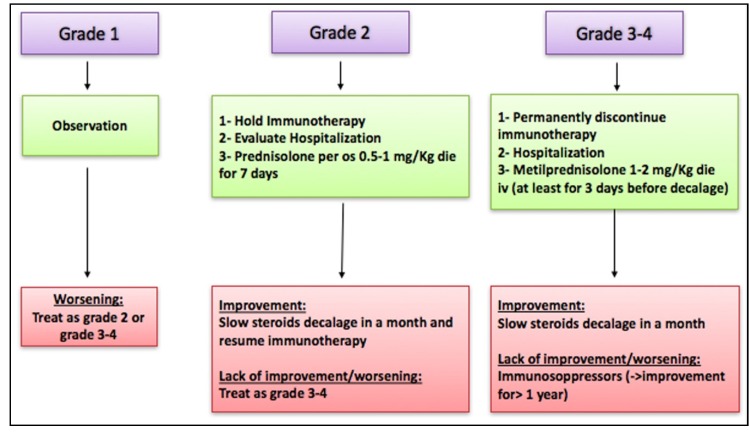
Neurological immune-related adverse events (irAEs) management guidelines.

**Table 1 cancers-11-00296-t001:** Selection of reported cases of neurological immune-related Adverse Events (irAEs) due to anti PD-1 immunotherapy.

Reference	Immuno-Therapy/Administered Cycles	irAE	Symoptoms	Exams	Treatment	Responce
Kao JC et al. 2017 [6]	Pembro/11	Cerebellar ataxia	Cerebellar ataxia and dysarthria	Not specified	Stop Immunotherapy	Improvement
Kao JC et al. 2017 [6]	Nivo/14	Headache	Headache	Not specified	Dexamethasone (4 mg twice daily) for 1 week	Improvement
Hasegawa Y. et al. 2017 [10]	Nivo/2	Myasthenia Gravis	Left eyelid ptosis, dyspnea and muscle weakness	Grade 4 CPK elevation AchR positive	Prednisolone Plasmapheresis and intravenous immune globulin	Improvement
Blackmon J. et al. 2016 [11]	Nivo/14	Encephalitis limbic	Unspecified	MRI abnormal	Steroids and Stop immunotherapy	No Improvement
Feng et al. 2017 [3]	Pembro/2	Encephalopaty	Somnolence, confusion, ataxia, expressive dysphasia, and cognitive impairment	CSF normal AChR normal MRI: Post-operative changes EEG: Generalized intermittent slowing consistent with a mild diffuse encephalopathy.	Methylprednisolone 1 g IV	Improvement
Polat et al. 2016 [12]	Nivo/3	Myasthenia Gravis	Bilateral ptosis and intermittent diplopia	AChR negative MRI: Negative	Stop Immunotherapy Pyridostigmine 45 mg every 6 h	Improvement
Sciacca et al. 2016 [13]	Nivo/3	Myasthenia Gravis	Bilateral ptosis, nasal speech, and proximal limb weakness	AChR positive EMG: Abnormal	Stop Immunotherapy Prednisone 50 mg	Improvement

Pembro: Pembrolizumab; MRI: magnetic resonance imaging; CSF: Cerebrospinal fluid; IV, intravenously; Nivo: nivolumab; PO, orally; Ig, immunoglobulin; AChRAb: Acetylcholine Receptor Autoantibody; EEG: Electroencephalogram; CPK: creatine phosphokinase.

**Table 2 cancers-11-00296-t002:** Symptoms, signs, and frequency of the principal neurological irAEs.

Neurological irAEs	Symptoms	Frequency Grade 1–2	Frequency Grade 3–4
Peripheral Polineuropathy	-Hypo/areflexia -Sensory ataxia -Numbness -Painful/Painless paresthesia -Central neurologic abnormalities (i.e., cranial nerve palsies)	5%	6%
Myasthenia Gravis	-Weakness of eyes’ muscles (ptosis and strabismus) -Dysarthria -Dysphagia -Dysphonia -Drooping head -Facial paralysis -Difficulty to lift objects or to climb stairs -Respiratory failure	6–12%	0.2–0.4%
Bell’s Palsy	-Drooping eye -Loss of nasolabial fold -Drooping corner of mouth -Dysgeusia -Neck, mastoid or ear pain -Hyperacusis or altered facial sensation	NA	NA
Encephalitis	-Confusion -Altered behavior -Headaches -Seizures -Short-term memory loss -Depressed levels of consciousness -Focal weakness -Speech abnormalities	1–3%	1%

NA: Not applicable.

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
