# Peer review of "PD-1 Inhibitors-Related Neurological Toxicities in Patients with Non-Small-Cell Lung Cancer: A Literature Review"

_cancers, 2019, doi:10.3390/cancers11030296_

Round 1

Reviewer 1 Report

The manuscript by Aurora et al 2019 provides an insight into the range of neurological symptoms experienced by patients undergoing PD-1 inhibitor treatment for non-small-cell lung cancer.  The paper is of interest, though personally I would suggest an editorial decision is required as to whether the article is suitable for publication in Cancers or would better suit a Clinical Investigations based journal.  This suggestion is based upon the inclusion of clinical cases, and a brief literature review of other clinical symptomology and treatments associated with neurological disorders, rather than a paper focussed upon cancer itself.

A limitation of the article is that it is also only a brief literature review, and it would be more beneficial of the authors to produce a systematic review, and it possible an associated meta-analysis.  From a more comprehensive review, the authors may be able to further consider clinical guidelines associated with patient management.

The manuscript itself requires an extensive polish of the English, as it is replete with grammatical and typographical errors.

Author Response

1-The manuscript by Aurora et al 2019 provides an insight into the range of neurological symptoms experienced by patients undergoing PD-1 inhibitor treatment for non-small-cell lung cancer.  The paper is of interest, though personally I would suggest an editorial decision is required as to whether the article is suitable for publication in Cancers or would better suit a Clinical Investigations based journal. 

Thanks for your interest in our paper.

2- A limitation of the article is that it is also only a brief literature review, and it would be more beneficial of the authors to produce a systematic review, and it possible an associated meta-analysis. 

Thanks for your major suggestion. We did a systematic review of anti PDL-1 related neurological adverse events  in NSCLC.

Nevertheless, we feel that metanalysis on the identified papers is not feasible due to their high heterogeneity and the small number of cases reported in literature. In line with your remark and for the same of clarify, we have included a table which summarizes a selection of cases of neurological irAEs due to anti PD-1 immunotherapy.

3- From a more comprehensive review, the authors may be able to further consider clinical guidelines associated with patient management.

We have now added this information to the text

4- The manuscript itself requires an extensive polish of the English, as it is replete with grammatical and typographical errors.

The paper now has been read by a Native English speaker (changes are tracked).

Reviewer 2 Report

The reviewer consider that this manuscript is not a literature review but a case report. Four cases who showed neurological irAEs after immunotherapy are shown, and discussion (together with data of previous publications) are added for each four disorder. However, as a case report, the presented data is not enough. For example, the 1st-line regimen is not described in Case 1. No reason is shown about the death of this patient, although the authors described that there was a benefit on neurological symptoms. No data about AchR antibody level, in Case 2, after steroid therapy is not shown. The authors did not describe how much improvement was seen for the neurological symptoms and transaminase values. Additional comments are as follows.

1. Figure 1. The authors should describe the souse of this Figure (what guideline did the authors refer to?)

2. The reviewer think Figure 2-5 are not necessary. These figures just indicate symptoms related to neurological disorders. In addition, Figure 2 and Figure 5 are completely the same.

3. There may be no reason that they exclude neurological irAE data for anti-PD-L1 drugs and IO combination.

Author Response

1-The reviewer consider that this manuscript is not a literature review but a case report. Four cases who showed neurological irAEs after immunotherapy are shown, and discussion (together with data of previous publications) are added for each four disorder.

Thanks for your major suggestion. However, since the neurological toxicity is described as a rare adverse event, we thought to start from clinical cases to better explain how difficult could be to reach a prompt diagnosis, adequate management and treatments, and the importance of a multidisciplinary approach.

2-However, as a case report, the presented data is not enough. For example, the 1st-line regimen is not described in Case 1.

We have now added this information to the text.

3-No reason is shown about the death of this patient, although the authors described that there was a benefit on neurological symptoms.

The patient did not die, we now tried to better explained the case.

4-No data about AchR antibody level, in Case 2, after steroid therapy is not shown.

We have now added this information to the text.

5-The authors did not describe how much improvement was seen for the neurological symptoms and transaminase values.

We have now added this information to the text.

6- Figure 1. The authors should describe the souse of this Figure (what guideline did the authors refer to?) We have now added this information to the text.

7- The reviewer think Figure 2-5 are not necessary.

The journal authors’ instructions require to add at least 2 figures or tables, this is the reason why we decided to add these figures..

8-Figure 2 and Figure 5 are completely the same.

We have now corrected the problem

9-There may be no reason that they exclude neurological irAE data for anti-PD-L1 drugs and IO combination.

We thank you for your major suggestion but we chose neurological irAE in NSCLC patients treated with Nivolumab and Pembrolizumab as the target of our report, because actually they are the widest and most frequent immunotherapies used, in face of a lack of sufficient data on neurological adverse event, in this setting.

Round 2

Reviewer 2 Report

The reviewer think that this manuscript is improved by the revision. However, the Figures 2-5 are not necessary, and I cannot understand the reason for these figures (the authors responded to my previous comments saying that the journal author’s instruction requested at least 2 figures or tables and this is the reason for the figures). The reviewer suggests summarizing Figure 2 -5 into one table, showing the symptoms related to each neurological irAEs. I also suggest adding the frequencies of each symptoms in each neurological disorder. If there is not enough data about the frequency of symptoms, the authors can use the frequency of symptoms for respective autoimmune disease. Such data will help clinicians to diagnose neurological irAEs.

In addition, the reviewer suggests adding treatment duration of pembrolizumab or nivolumab in Table 1.

Author Response

1-    The reviewer suggests summarizing Figure 2 -5 into one table, showing the symptoms related to each neurological irAEs. I also suggest adding the frequencies of each symptoms in each neurological disorder. If there is not enough data about the frequency of symptoms, the authors can use the frequency of symptoms for respective autoimmune disease. Such data will help clinicians to diagnose neurological irAEs.

We thank you for your suggestions. We have now added these informations creating a table as requested

2-     In addition, the reviewer suggests adding treatment duration of pembrolizumab or nivolumab in Table 1.

We have now added these informations